# Conformal Isometry of Lie Group Representation in Recurrent Network of Grid Cells

**Dehong Xu**[*]                                             XUDEHONG1996@UCLA.EDU
*Department of Statistics, UCLA*

**Ruiqi Gao**[*]                                             RUIQIG@GOOGLE.COM
*Google Research, Brain Team*

**Wen-Hao Zhang**                           WENHAO.ZHANG@UTSOUTHWESTERN.EDU
*Lyda Hill Department of Bioinformatics and O'Donell Brain Institute, UT Southwestern Medical Center*

**Xue-Xin Wei**                                             WEIXX@UTEXAS.EDU
*Departments of Neuroscience and Psychology, Center for Perceptual Systems, Center for Theoretical and Computational Neuroscience, UT Austin*

**Ying Nian Wu**                                             YWU@STAT.UCLA.EDU
*Department of Statistics, UCLA*

**Editors:** Sophia Sanborn, Christian Shewmake, Simone Azeglio, Arianna Di Bernardo, Nina Miolane

## Abstract

The activity of the grid cell population in the medial entorhinal cortex (MEC) of the mammalian brain forms a vector representation of the self-position of the animal. Recurrent neural networks have been proposed to explain the properties of the grid cells by updating the neural activity vector based on the velocity input of the animal. In doing so, the grid cell system effectively performs path integration. In this paper, we investigate the algebraic, geometric, and topological properties of grid cells using recurrent network models. Algebraically, we study the Lie group and Lie algebra of the recurrent transformation as a representation of self-motion. Geometrically, we study the conformal isometry of the Lie group representation where the local displacement of the activity vector in the neural space is proportional to the local displacement of the agent in the 2D physical space. Topologically, the compact and connected abelian Lie group representation automatically leads to the torus topology commonly assumed and observed in neuroscience. We then focus on a simple non-linear recurrent model that underlies the continuous attractor neural networks of grid cells. Our numerical experiments show that conformal isometry leads to hexagon periodic patterns in the grid cell responses and our model is capable of accurate path integration. Code is available at https://github.com/DehongXu/grid-cell-rnn.

**Keywords:** Grid cells, Conformal isometry, Lie group representation, Lie algebra, Peter-Weyl theory, Flat torus.

---

[*] Equal contribution

## 1. Introduction

Grid cells (Hafting et al., 2005; Fyhn et al., 2008; Yartsev et al., 2011; Killian et al., 2012; Jacobs et al., 2013; Doeller et al., 2010) in the mammalian dorsal medial entorhinal cortex (MEC) exhibit striking hexagon grid patterns when the agent (e.g., a rodent) navigates in 2D open environments  (Fyhn et al., 2004; Hafting et al., 2005; Fuhs and Touretzky, 2006; Burak and Fiete, 2009; Sreenivasan and Fiete, 2011; Blair et al., 2007; Couey et al., 2013; de Almeida et al., 2009; Pastoll et al., 2013; Agmon and Burak, 2020). It has been hypothesized that grid cell system performs path integration  (Darwin, 1873; Etienne and Jeffery, 2004; Hafting et al., 2005; Fiete et al., 2008; McNaughton et al., 2006; Gil et al., 2018; Ridler et al., 2019; Horner et al., 2016). That is, the grid cells integrate the self-motion of the animal over time to keep track of the animal's own location in space. This can be implemented by a recurrent neural network that takes the velocity of the self-motion as input, and transforms the activities of the grid cells based on the velocity inputs. The animal's self-position can then be decoded from the activities of the grid cells.

Collectively, the activities of the grid cell population form a vector in the high-dimensional neural activity space. This provides a representation of the self-position of the agent in space. The recurrent network transforms the activity vector based on the movement velocity of the agent, so that the transformation is a representation of self-motion, when considered from the perspective of representational learning. The vector and the transformation together form a representation of the 2D Euclidean group, which is an abelian additive Lie group.

In a recent paper, Gao et al. (2021) studied the group representation property and the isotropic scaling or conformal isometry property for the general transformation model. In the context of linear transformation models, they connected this property to the hexagon periodic patterns of the grid cell response maps. With the conformal isometry property of the transformation of the recurrent neural network, the change of the activity vector in the neural space is proportional to the input velocity of the self-motion in the 2D physical space. Although Gao et al. (2021) studied general transformation model theoretically, they focused on a prototype model of linear recurrent network numerically, which has an explicit algebraic and geometric structure in the form of a matrix group of rotations.

In this paper, we study conformal isometry in the context of the non-linear recurrent model that underlies the hand-crafted continuous attractor neural network (CANN) (Burak and Fiete, 2009; Couey et al., 2013; Pastoll et al., 2013; Agmon and Burak, 2020).  In particular, we will focus on the vanilla version of the recurrent network that is linear in the vector representation of self-position and is additive in the input velocity, followed by an element-wise non-linear rectification (such as ReLU). This model has the simplicity that it is additive in input velocity before rectification. We also explore more complex variants for non-linear recurrent networks, such as the long short-term memory network (LSTM) (Hochreiter and Schmidhuber, 1997). Such models have been studied in recent works (Cueva and Wei, 2018; Banino et al., 2018; Sorscher et al., 2019; Cueva et al., 2020).

Our numerical experiments show that our conformal isometry condition is able to learn highly structured multi-scale hexagon grid code, consistent with the properties of experimentally observed grid cells of rodents. In addition, our learned model is capable of accurate path integration over a long distance. Our results generalize previous results of linear network

models in Gao et al. (2019, 2021) to an important class of non-linear neural network models in theoretical neuroscience that are more physiologically realistic.

## 2. Lie group representation and conformal isometry

### 2.1. Representations of self-position and self-motion

We start by introducing the basic components of our model. $\boldsymbol{x} = (x_1, x_2) \in \mathbb{R}^2$ denotes the agent's position. Let $\Delta\boldsymbol{x} = (\Delta x_1, \Delta x_2)$ be the input velocity of the self-motion, i.e., displacement of the agent within a unit time, after which the agent moves from $\boldsymbol{x}$ to $\boldsymbol{x} + \Delta\boldsymbol{x}$.

We assume $\boldsymbol{v}(\boldsymbol{x}) = (v_i(\boldsymbol{x}), i = 1, ..., D)$ to be the vector representation of self-position $\boldsymbol{x}$, where each element $v_i(\boldsymbol{x})$ can be interpreted as the activity of a grid cell when the agent is at position $\boldsymbol{x}$. $(v_i(\boldsymbol{x}), \forall \boldsymbol{x})$ corresponds to the response map of grid cell $i$. $D$ is the dimensionality of $\boldsymbol{v}$, i.e., the number of grid cells. We refer to the space of $\boldsymbol{v}$ as the "neural space". We normalize $\|\boldsymbol{v}(\boldsymbol{x})\| = 1$ in our experiments.

The set $(\boldsymbol{v}(\boldsymbol{x}), \boldsymbol{x} \in \mathbb{R}^2)$ forms a 2D manifold, or an embedding of $\mathbb{R}^2$, in the $D$-dimensional neural space. We will refer to $(\boldsymbol{v}(\boldsymbol{x}), \boldsymbol{x} \in \mathbb{R}^2)$ as the "coding manifold".

With self-motion $\Delta\boldsymbol{x}$, the vector representation $\boldsymbol{v}(\boldsymbol{x})$ is transformed to $\boldsymbol{v}(\boldsymbol{x} + \Delta\boldsymbol{x})$ by a general transformation model:

$$\boldsymbol{v}(\boldsymbol{x} + \Delta\boldsymbol{x}) = F(\boldsymbol{v}(\boldsymbol{x}), \Delta\boldsymbol{x}) = F_{\Delta\boldsymbol{x}}(\boldsymbol{v}(\boldsymbol{x})), \tag{1}$$

where by simplifying $F(\cdot, \Delta\boldsymbol{x})$ as $F_{\Delta\boldsymbol{x}}(\cdot)$ in notation, we emphasize that the transformation $F$ is dependent on $\Delta\boldsymbol{x}$. While $\boldsymbol{v}(\boldsymbol{x})$ is a representation of $\boldsymbol{x}$, $F_{\Delta\boldsymbol{x}}$ is a representation of $\Delta\boldsymbol{x}$. $(\boldsymbol{v}(\boldsymbol{x}), \forall \boldsymbol{x})$ and $(F_{\Delta\boldsymbol{x}}(\cdot), \forall \Delta\boldsymbol{x})$ together form a representation of the 2D additive Euclidean group $\mathbb{R}^2$, which is an abelian Lie group. Specifically, we have the following group representation condition for the transformation model:

**Condition 1.** *(Algebraic condition on Lie group representation). For any $\boldsymbol{x}$, we have (1) $F_0(\boldsymbol{v}(\boldsymbol{x})) = \boldsymbol{v}(\boldsymbol{x})$, and (2) $F_{\Delta\boldsymbol{x}_1 + \Delta\boldsymbol{x}_2}(\boldsymbol{v}(\boldsymbol{x})) = F_{\Delta\boldsymbol{x}_2}(F_{\Delta\boldsymbol{x}_1}(\boldsymbol{v}(\boldsymbol{x}))) = F_{\Delta\boldsymbol{x}_1}(F_{\Delta\boldsymbol{x}_2}(\boldsymbol{v}(\boldsymbol{x})))$ for any $\Delta\boldsymbol{x}_1$ and $\Delta\boldsymbol{x}_2$.*

Condition 1(1) requires that the coding manifold $(\boldsymbol{v}(\boldsymbol{x}), \forall \boldsymbol{x})$ are fixed points of $F_0$ with $\Delta\boldsymbol{x} = 0$. If $F_0$ is further a contraction off the coding manifold, then $(\boldsymbol{v}(\boldsymbol{x}), \forall \boldsymbol{x})$ are the attractor points of $F_0$. Condition 1(2) requires that moving in one step with displacement $\Delta\boldsymbol{x}_1 + \Delta\boldsymbol{x}_2$ should be the same as moving in two steps with displacements $\Delta\boldsymbol{x}_1$ and $\Delta\boldsymbol{x}_2$ respectively. The group representation condition is the necessary condition for any valid transformation model (Equation (1)) of grid cells.

Group representation is a central theme in modern mathematics and physics (Zee, 2016). However, most of the transformations studied in mathematics and physics are linear transformations that form matrix groups, and the coding manifold $(\boldsymbol{v}(\boldsymbol{x}), \forall \boldsymbol{x})$ is often made implicit. Gao et al. (2019) focused on matrix groups, with $F_{\Delta\boldsymbol{x}}(\boldsymbol{v}(\boldsymbol{x})) = \boldsymbol{M}(\Delta\boldsymbol{x})\boldsymbol{v}(\boldsymbol{x})$, so that $\boldsymbol{M}(\Delta\boldsymbol{x}_1 + \Delta\boldsymbol{x}_2)\boldsymbol{v}(\boldsymbol{x}) = \boldsymbol{M}(\Delta\boldsymbol{x}_1)\boldsymbol{M}(\Delta\boldsymbol{x}_2)\boldsymbol{v}(\boldsymbol{x}) = \boldsymbol{M}(\Delta\boldsymbol{x}_2)\boldsymbol{M}(\Delta\boldsymbol{x}_1)\boldsymbol{v}(\boldsymbol{x})$. Gao et al. (2021) studied general transformation model theoretically, but then focused on the linear transformation model in their numerical experiments. Since the transformations in RNN are usually non-linear, we will focus on non-linear transformation models in this paper.

## 2.2. Conformal embedding and conformal isometry

For an infinitesimal self-motion $\delta\boldsymbol{x}$, it is straightforward to derive a first-order Taylor expansion of the transformation model in Equation (1) with respect to $\delta\boldsymbol{x}$

$$\begin{aligned}
\boldsymbol{v}(\boldsymbol{x}+\delta\boldsymbol{x}) &= F_{\boldsymbol{0}}(\boldsymbol{v}(\boldsymbol{x})) + F_{\boldsymbol{0}}'(\boldsymbol{v}(\boldsymbol{x}))\delta\boldsymbol{x} + o(|\delta\boldsymbol{x}|) \\
&= \boldsymbol{v}(\boldsymbol{x}) + f(\boldsymbol{v}(\boldsymbol{x}))\delta\boldsymbol{x} + o(|\delta\boldsymbol{x}|),
\end{aligned} \tag{2}$$

where $f(\boldsymbol{v}(\boldsymbol{x})) = \frac{\partial F_{\Delta\boldsymbol{x}}}{\partial\Delta\boldsymbol{x}^{\top}}(\boldsymbol{v}(\boldsymbol{x}))\mid_{\Delta\boldsymbol{x}=0}$ is a $D \times 2$ matrix.

While $(F_{\Delta\boldsymbol{x}}, \forall\Delta\boldsymbol{x} \in \mathbb{R}^2)$ forms an abelian Lie group, its derivative of $\Delta\boldsymbol{x}$ at 0, i.e., $f$, spans its Lie algebra. Both $F_{\Delta\boldsymbol{x}}$ and $f$ are transformations acting on the coding manifold $(\boldsymbol{v}(\boldsymbol{x}), \forall\boldsymbol{x})$.

We identify the conformal isometry condition of the Lie group representation as follows:

**Condition 2.** *(Geometric condition on conformal embedding and conformal isometry).*

$$f(\boldsymbol{v}(\boldsymbol{x}))^{\top} f(\boldsymbol{v}(\boldsymbol{x})) = s^2 \boldsymbol{I}_2, \ \forall\boldsymbol{x}, \tag{3}$$

*where $\boldsymbol{I}_2$ is the 2-dimensional identity matrix. That is, the two column vectors of $f(\boldsymbol{v}(\boldsymbol{x}))$ are of equal norm $s$, and are orthogonal to each other.*

Under the condition above, $\boldsymbol{v}(\boldsymbol{x}+\delta\boldsymbol{x}) - \boldsymbol{v}(\boldsymbol{x}) \approx f(\boldsymbol{v}(\boldsymbol{x}))\delta\boldsymbol{x}$ is conformal to $\delta\boldsymbol{x}$, i.e., the 2D local Euclidean space of $(\delta\boldsymbol{x})$ in the physical space is embedded conformally as another 2D local Euclidean space $f(\boldsymbol{v}(\boldsymbol{x}))\delta\boldsymbol{x}$ in the neural activity space. We only need to replace the two orthogonal axes for $\delta\boldsymbol{x}$ in the 2D physical space by the two column vectors of $f(\boldsymbol{v}(\boldsymbol{x}))$ in the neural activity space.

An equivalent statement for the above condition is

$$\|\boldsymbol{v}(\boldsymbol{x}+\delta\boldsymbol{x}) - \boldsymbol{v}(\boldsymbol{x})\| = s\|\delta\boldsymbol{x}\| + o(\|\delta\boldsymbol{x}\|), \forall\boldsymbol{x}, \delta\boldsymbol{x}. \tag{4}$$

That is, the displacement in the neural space is proportional to that in the 2D physical space.

Note that since our analysis is local, $s$ may depend on $\boldsymbol{x}$. If $s$ is a global constant, then the coding manifold $(\boldsymbol{v}(\boldsymbol{x}), \forall\boldsymbol{x})$ has a flat intrinsic geometry (imagining folding a piece of paper without stretching it).

## 2.3. 2D torus, 2D periodicity, and hexagon grid patterns

The 2D torus topology is commonly assumed *a priori* in the continuous attractor neural networks (CANN) for grid cells (Burak and Fiete, 2009; Couey et al., 2013; Pastoll et al., 2013; Agmon and Burak, 2020). The torus topology has been recently supported by analyzing data from population of simultaneously recorded grid cells (Gardner et al., 2022). Within our framework, we find that such a topology is in fact a theoretical consequence of the group representation condition (Condition 1) due to a theorem in Lie group theory.

Specifically, since the elements of $\boldsymbol{v}(\boldsymbol{x})$ represent biologically bounded neuron activities, the coding manifold $(\boldsymbol{v}(\boldsymbol{x}), \forall\boldsymbol{x})$ is bounded and compact, and therefore the group of $(F_{\Delta\boldsymbol{x}}, \forall\Delta\boldsymbol{x})$ is also compact. The Lie group $(F_{\Delta\boldsymbol{x}}, \forall\Delta\boldsymbol{x})$ is also connected since the 2D environment is connected. According to Lie group theory (Dwyer and Wilkerson, 1998), a compact and connected abelian Lie group $(F_{\Delta\boldsymbol{x}}, \forall\Delta\boldsymbol{x})$ has a topology of 2D torus, i.e., it is isomorphic to $\mathbb{S}_1 \times \mathbb{S}_1$, where $\mathbb{S}_1$ is a circle. Thus $(\boldsymbol{v}(\boldsymbol{x}) = F_{\boldsymbol{x}}(\boldsymbol{v}(0)), \forall\boldsymbol{x} \in \mathbb{R}^2)$ also forms a 2D torus.

While the group representation condition only gives us an algebraic structure, the conformal isometry condition (Condition 2) further fixes the geometry. Under the conformal isometry condition, if we further assume that the scaling factor $s$ is a constant globally for all $\boldsymbol{x}$, then the intrinsic geometry of the coding manifold $(\boldsymbol{v}(\boldsymbol{x}), \forall \boldsymbol{x})$ remains Euclidean, and the coding manifold is a flat torus. That is, the coding manifold is not only isomorphic to $\mathbb{S}_1 \times \mathbb{S}_1$, but is also conformally isometric to $\mathbb{S}_1 \times \mathbb{S}_1$. While isomorphism is defined by mapping between two spaces, isometry concerns about the metric properties. This leads to the periodic pattern in $(\boldsymbol{v}(\boldsymbol{x}), \forall \boldsymbol{x})$ over $\boldsymbol{x}$.

According to the theory of 2D Bravais lattice for 2D periodic patterns (Ashcroft et al., 1976), we can find two primitive vectors $\Delta \boldsymbol{x}_1$ and $\Delta \boldsymbol{x}_2$, with $\|\Delta \boldsymbol{x}_1\| = \|\Delta \boldsymbol{x}_2\|$, and $\boldsymbol{v}(\boldsymbol{x} + k_1 \Delta \boldsymbol{x}_1 + k_2 \Delta \boldsymbol{x}_2) = \boldsymbol{v}(\boldsymbol{x})$ for arbitrary integers $k_1$ and $k_2$. Along each primitive vector, for each period, $(\boldsymbol{v}(\boldsymbol{x} + c \Delta \boldsymbol{x}_i), c \in [0,1])$ traces out a circle in the neural space for $i = 1, 2$, causing the periodicity in $\boldsymbol{v}(\boldsymbol{x})$. According to the theory of 2D Bravais lattice, the angle between $\Delta \boldsymbol{x}_1$ and $\Delta \boldsymbol{x}_2$ can either be $\pi/2$ for square lattice or $2\pi/3$ for hexagon lattice. It is likely that the hexagon periodicity provides a better fit to place cells, in that the hexagon lattice provides denser packing of discrete Fourier components. While this seems to be intuitive, currently we have not been able to prove this point rigorously. It seems that hexagonal periodicity emerges under the general conditions of group representation and conformal isometry, independent of a specific form of the transformation model, as we observe empirically in our numerical experiments (Section 5).

## 3. Non-linear recurrent neural network

### 3.1. Model

In this paper, we mainly focus on studying transformations $F_{\Delta \boldsymbol{x}}(\cdot)$ that are locally approximated by non-linear recurrent neural networks, as studied in recent work (Cueva and Wei, 2018; Banino et al., 2018; Sorscher et al., 2019; Cueva et al., 2020). We start by assuming the following vanilla version of a non-linear recurrent network:

$$\boldsymbol{v}(\boldsymbol{x} + \Delta \boldsymbol{x}) = \text{ReLU}(\boldsymbol{W} \boldsymbol{v}(\boldsymbol{x}) + \boldsymbol{U} \Delta \boldsymbol{x}), \tag{5}$$

where $\text{ReLU}(a) = \max(0, a)$ is applied element-wise, $\boldsymbol{W}$ is a $D \times D$ weight matrix of recurrent connections, $\boldsymbol{U}$ is a $D \times 2$ matrix, and the self-motion $\Delta \boldsymbol{x} = (\Delta x_1, \Delta x_2)^\top$ is treated as $2 \times 1$ vector. Note that the above model is an accurate approximation to $F_{\Delta \boldsymbol{x}}(\cdot)$ only for small $\Delta \boldsymbol{x}$, as it may not satisfy Condition 1 in general for large $\Delta \boldsymbol{x}$. Following Condition 1(1), for $\Delta \boldsymbol{x} = 0$ we have $\boldsymbol{v}(\boldsymbol{x}) = \text{ReLU}(\boldsymbol{W} \boldsymbol{v}(\boldsymbol{x}))$. That is, the coding manifold $(\boldsymbol{v}(\boldsymbol{x}), \forall \boldsymbol{x})$ consists of the fixed points of $F_0(\cdot)$.

Compared to the matrix group in Gao et al. (2019) where $F_{\Delta \boldsymbol{x}}(\boldsymbol{v}(\boldsymbol{x})) = \boldsymbol{M}(\Delta \boldsymbol{x}) \boldsymbol{v}(\boldsymbol{x})$, and $\boldsymbol{M}(\Delta \boldsymbol{x})$ is further derived in Gao et al. (2021) as an exponential map that is highly non-linear in $\Delta \boldsymbol{x}$, the above model (5) is much simpler and more biologically plausible. Before ReLU, we have a single recurrent weight matrix $\boldsymbol{W}$ that is independent of $\Delta \boldsymbol{x}$, and $\Delta \boldsymbol{x}$ enters the equation additively. The ReLU rectification plays a critical role for the overall non-linear effect of $\Delta \boldsymbol{x}$. The non-linear rectification in neural networks makes the transformations much more expressive than matrix representations in modern mathematics and physics.

For this model, we can derive $f(\boldsymbol{v}(\boldsymbol{x}))$ as

$$f(\boldsymbol{v}(\boldsymbol{x})) = \mathbf{1}(\boldsymbol{W} \boldsymbol{v}(\boldsymbol{x}) > \mathbf{0}) \odot \boldsymbol{U}, \tag{6}$$

where $\mathbf{1}(\cdot)$ is a vector of binary indicators calculated element-wise, and $\odot$ is row-wise product. The indicator vector $\mathbf{1}(\boldsymbol{W}\boldsymbol{v}(\boldsymbol{x}) > \boldsymbol{0})$ changes as $\boldsymbol{x}$ changes, and it controls the change of $\boldsymbol{v}$ in the neural space according to $\Delta\boldsymbol{x}$.

Since $\mathbf{1}(\cdot)$ is not differentiable, we propose to define the loss function to enforce the conformal isometry condition based on the equivalent statement in Equation (4), as discussed in Section 4.3. In Appendix C, we also discuss a possible mechanism where conform isometry can be automatically satisfied by design.

**Modules.** Since biological grid cells are organized in discrete modules with different spatial scales (Stensola et al., 2012; Barry et al., 2007), our model assumes that the vector representations $\boldsymbol{v}(\boldsymbol{x})$ are divided into sub-vectors analogous to modules. Accordingly, the transformation should also be module-wise: $\boldsymbol{W}$ is block-diagonal and $\boldsymbol{U}$ is divided into sub-blocks by row. To address the point that the module-wise transformation construction is just to keep the consistency with isometry assumption but not the reason for the emergence of hexagonal periodicity, we have performed an ablation study (see Appendix A.3).

**More complex transformations.** For local transformation models, we also explore the Long Short-Term Memory network (LSTM) (Hochreiter and Schmidhuber, 1997), a more complex variant of recurrent networks. We observed the hexagonal grid patterns in both types of transformation models: empirically, hexagonal periodicity under the conformal isometry condition is not specific to the particular form of the transformation model.

### 3.2. Eigen analysis

Next, we will deepen our theoretical understanding of the model (5) by conducting eigen analysis under the conformal isometry condition.

**Theorem 1.** *Under conformal isometry condition, for every $\boldsymbol{x}$, the two columns of $\mathbf{1}(\boldsymbol{W}\boldsymbol{v}(\boldsymbol{x}) > \boldsymbol{0}) \odot \boldsymbol{U}$ are orthogonal, and they are eigenvectors of $\mathbf{1}(\boldsymbol{W}\boldsymbol{v}(\boldsymbol{x}) > \boldsymbol{0}) \odot \boldsymbol{W}$ with eigenvalue 1.*

See Appendix B for proof. If the magnitudes of all the other eigenvalues of $\mathbf{1}(\boldsymbol{W}\boldsymbol{v}(\boldsymbol{x}) > \boldsymbol{0}) \odot \boldsymbol{W}$ are less than 1, the recurrent network is a contraction off the coding manifold, which is similar to the eigen structures of general continuous attractor neural networks as shown in Fung et al. (2010).

## 4. Reconstructing place cells and learning by numerical optimization

### 4.1. Place cells and decoding

For open fields, we model place cells (O'Keefe, 1979) by Gaussian kernels and connect them to grid cells by the basis expansion model (Dordek et al., 2016; Sorscher et al., 2019):

$$A(\boldsymbol{x}, \boldsymbol{p}) = \exp(-\|\boldsymbol{x} - \boldsymbol{p}\|^2/2\sigma^2) = \langle \boldsymbol{v}(\boldsymbol{x}), \boldsymbol{q}(\boldsymbol{p}) \rangle, \tag{7}$$

where $A(\boldsymbol{x}, \boldsymbol{p})$ represent the place field centered at position $\boldsymbol{p}$. $A(\boldsymbol{x}, \boldsymbol{p})$ measures the adjacency of $\boldsymbol{x}$ to $\boldsymbol{p}$. $\boldsymbol{q}(\boldsymbol{p})$ is the query vector of place cell $\boldsymbol{p}$, which can be interpreted as connection weights between the grid cells and place cell $\boldsymbol{p}$. For vector $\boldsymbol{v}$, we can decode its position $\hat{\boldsymbol{x}}$ by

$$\hat{\boldsymbol{x}} = \arg\max_{\boldsymbol{p}} \langle \boldsymbol{v}, \boldsymbol{q}(\boldsymbol{p}) \rangle, \tag{8}$$

i.e., we choose the position $\boldsymbol{p}$ that is closest to the position encoded by $\boldsymbol{v}$. In our experiments, we learn the query vectors $(\boldsymbol{q}(\boldsymbol{p}), \forall \boldsymbol{p})$ together with $(\boldsymbol{v}(\boldsymbol{x}), \forall \boldsymbol{x})$, $\boldsymbol{W}, \boldsymbol{U}$.

### 4.2. Peter-Weyl theory

In the model defined by Equation (7), $\boldsymbol{v}(\boldsymbol{x})$ generated by the transformation model serves as a set of basis functions to reconstruct the set of functions $A(\boldsymbol{x}, \boldsymbol{p}), \forall \boldsymbol{p}$. This is related to the Peter-Weyl theory (Taylor, 2002), which generalizes the Fourier analysis to Lie group and shows that the basis functions arise from Lie group representation. $\boldsymbol{v}(\boldsymbol{x})$ can be used to reconstruct and interpolate value functions of $\boldsymbol{x}$ in general. Peter-Weyl theory is about matrix Lie groups and irreducible representations. In our work, we focus on non-linear transformation groups which can still generate basis functions.

Peter-Weyl theory naturally connects two roles of grid cells: (1) path integration, and (2) basis expansion. It can be interesting to generalize Peter-Weyl theory to non-linear transformations.

### 4.3. Loss function

Assuming the kernel $A(\boldsymbol{x}, \boldsymbol{p})$ is given as in Equation (7), we can learn the model by minimizing the following loss term:

$$L_1 = \sum_{t=1}^{T} \sum_{\boldsymbol{p}} \mathbb{E}_{\boldsymbol{x}, \Delta \boldsymbol{x}} [A(\boldsymbol{x} + \Delta \boldsymbol{x}_1 + ... + \Delta \boldsymbol{x}_t, \boldsymbol{p}) - \langle F_{\Delta \boldsymbol{x}_t} ... F_{\Delta \boldsymbol{x}_1}(\boldsymbol{v}(\boldsymbol{x})), \boldsymbol{q}(\boldsymbol{p}) \rangle]^2. \quad (9)$$

The learnable parameters includes $(\boldsymbol{v}(\boldsymbol{x}), \forall \boldsymbol{x})$, $(\boldsymbol{q}(\boldsymbol{p}), \forall \boldsymbol{p})$, and parameters in $F_{\Delta \boldsymbol{x}}$. The expectations are estimated by Monte Carlo samples from simulated trajectories. $A(\boldsymbol{x}, \boldsymbol{p})$ are Gaussian kernels with predefined $\sigma$. In practice, we add an additional zero-step version of $L_1$, i.e. the expectation term changes to $[A(\boldsymbol{x}, \boldsymbol{p}) - \langle \boldsymbol{v}(\boldsymbol{x}), \boldsymbol{q}(\boldsymbol{p}) \rangle]^2$.

To ensure that the conformal isometry condition is satisfied, we add an extra loss term based on the equivalent statement of conformal isometry. Following Equation (4), for simplicity, we first denote $\boldsymbol{s}(\boldsymbol{x}, \Delta \boldsymbol{x}) = (\|\boldsymbol{v}(\boldsymbol{x}) - \boldsymbol{v}(\boldsymbol{x} + \Delta \boldsymbol{x})\| / \|\Delta \boldsymbol{x}\|)^2$. Then we propose a conformal isometry loss:

$$L_2 = \mathbb{E}_{\boldsymbol{x}, \Delta \boldsymbol{x}_1, \Delta \boldsymbol{x}_2} [\boldsymbol{s}(\boldsymbol{x}, \Delta \boldsymbol{x}_1) - \boldsymbol{s}(\boldsymbol{x}, \Delta \boldsymbol{x}_2)]^2, \quad (10)$$

where $\Delta \boldsymbol{x}_1$ and $\Delta \boldsymbol{x}_2$ are sampled such that they have the same length $\Delta r$ but with different directions, i.e. $\Delta \boldsymbol{x}_1 = (\Delta \boldsymbol{r} \cos \theta_1, \Delta \boldsymbol{r} \sin \theta_1)$, $\Delta \boldsymbol{x}_2 = (\Delta \boldsymbol{r} \cos \theta_2, \Delta \boldsymbol{r} \sin \theta_2)$. Moreover, we add another regularization term to penalize $\|\boldsymbol{q}(\boldsymbol{p})\|^2$.

## 5. Experiments

We optimized the model using simulated trajectories as training data. The environment was assumed to be a 1m × 1m squared open field, discretized into a 40 × 40 lattice. $\boldsymbol{v}(\boldsymbol{x})$ is of 1800 dimensions, which was partitioned into 150 modules with module size 12. For $A(\boldsymbol{x}, \boldsymbol{p})$, we used a Gaussian adjacency kernel with $\sigma = 0.07$. We trained a 10-step recurrent network as the transformation model, i.e., $T = 10$ in the loss term $L_1$.

For $L_1$, the displacement of $\Delta \boldsymbol{x}_t$ was restricted to be smaller than 3 grids. For the range of $\Delta r$ in $L_2$, we hypothesize that it can be proportional to the scale of the module (i.e., $s$ in Equation (4)), which is also reflected as the scale of the learned hexagon patterns. Thus we adaptively adjusted the upper bound of $\Delta r$ for each module during training, based on the scale of the learned hexagon patterns at the current training stage. We averaged the scales of the learned patterns within each module to represent the scale of that module. We set the upper bound of $\Delta r$ for the module with the largest average scale to be 15 grids. The ranges of $\Delta r$ for the remaining modules are adjusted according to their scales.

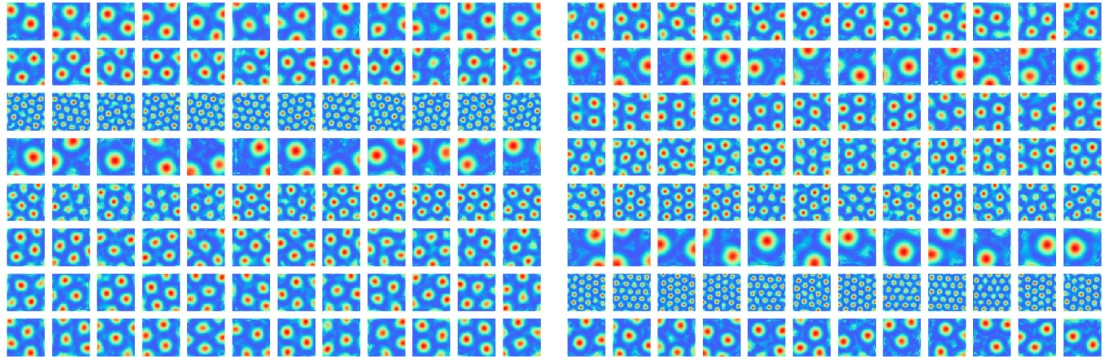

Figure 1: Hexagon grid firing patterns emerge in the learned $\boldsymbol{v}(\boldsymbol{x})$. Each row represents the firing patterns of all the cells in the same module. Each unit shows the learned neuron activity over the whole 2D squared environment. The figure shows patterns from 16 randomly selected modules.

## 5.1. Hexagon patterns

Figure 1 shows the learned firing patterns of $\boldsymbol{v}(\boldsymbol{x}) = (v_i(\boldsymbol{x}), i = 1, ..., d)$ over the $40 \times 40$ lattice of $\boldsymbol{x}$. We randomly selected 16 modules out of 150 modules for visualization purposes. Each image corresponds to the response map of a grid cell. Every row shows the learned units that belong to the same module. The hexagonal patterns in the emerging activity patterns are evident. We found that the loss term for imposing the conformal isometry was critical. Without it, the learned response maps showed stripe-like patterns (see Appendix A.3 for ablation results).

Table 1: Gridness scores and valid rates of grid cells of learned models. The first two lines are RNN models and the last two are LSTM models. Our models achieve higher gridenss score ($\uparrow$) and the percetage of valid grid cells ($\uparrow$) comparing to existing models.

| Model | Gridness score | % of grid cells |
|---|---|---|
| Sorscher et al. (2019) | 0.48 | 56.10 |
| Ours (RNN) | **0.77** | **72.5** |
| Banino et al. (2018) | 0.18 | 25.20 |
| Ours (LSTM) | **0.73** | **68.8** |

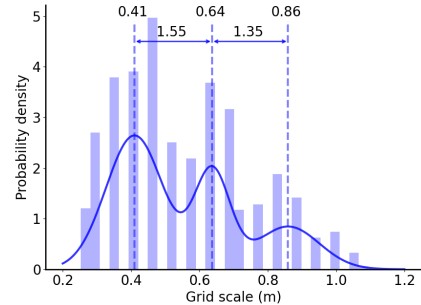

Figure 2: Histogram of grid scales of the learned model.

To quantitatively evaluate whether the learned patterns match regular hexagon grids, in Table 1, we report the gridness scores that are adopted from the literature of grid cells (Langston et al., 2010; Sargolini et al., 2006), as well as the valid percentage of grid cells with gridness score $> 0.37$ being the criteria. Figure 2 shows the histogram of the spatial scales of the learned hexagon patterns. The multi-modal distribution is fitted by a mixture of three Gaussians, which are centered at 0.41, 0.64 and 0.86 respectively. The ratios between adjacent centers are 1.55 and 1.35, which are in the range of the data from rodent grid cells (Stensola et al., 2012).

## 5.2. Path integration

We further evaluate whether the learned model is capable of accurate path integration. We perform path integration in two scenarios. First, for path integration with re-encoding, we decode $v_t \to x_t$ to physical space and then apply encoder $v(x_t) \to v'_t$ back to neuron space every few steps. This re-encoding strategy helps correct the errors accumulated in the neural space along the transformation. In the case without re-encoding, we apply transformation purely using neuron vector $v_t$. As shown in the left panel of Figure 3, the model can perform near exact path integration for 30 steps (short distance) without re-encoding. For long-distance path integration, we train a 20-step recurrent network model, and evaluate the model for 500 steps over 1000 trajectories. As shown in the right subfigure of Figure 3, if we re-encode every 20 steps, the path integration error for the last step is 0.028, while the average error over the 500 steps trajectory is 0.017. Without re-encoding, the error is relatively larger, where the average error is around 0.03 along the whole trajectory, and 0.08 for the last step.

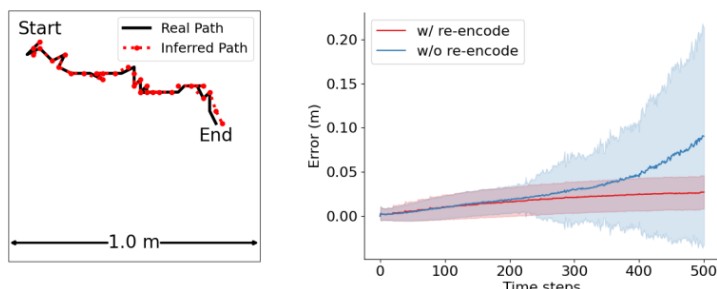

Figure 3: The learned model can perform accurate path integration. *Left*: path integration for 30 steps without re-encoding. Black: ground truth. Red: the inferred path by the learned model. *Right*: long distance (500-step) path integration error with (red) and without (blue) re-encoding by a learned 20-step RNN model over time steps. The average error and standard deviation are evaluated over 1000 trajectories.

To check if the model can apply precise encoding and decoding between physical space and neuron space, we also examine the fixed point condition by applying $v(x) \to v_t \to x'$. Ideally, the learned model can figure out the physical location $x_t$ purely from $v_t$. The $L_2$ error between $x$ and $x'$ is nearly zero ($< 0.005$).

### 5.3. Model with LSTM units

In this section, we evaluate the LSTM transformation model. The model is still trained by the same loss functions as the vanilla RNN model. Meanwhile, we force $q(p) > 0$ in training.

Examples of the learned patterns are visualized in Figure 4. Clear hexagon patterns are also evident. Different from the learned units from the vanilla recurrent network which are all non-negative, the learned $v(x)$ from the LSTM model can be either positive or negative, resulting in the color shift of the learned patterns. As shown in Table 1, the average gridness score for the LSTM model is 0.73, and 68.8% of the model units are classified as grid cells. We also evaluated the LSTM model on path integration still using 1000 trajectories and each trajectory is 500 steps long. With re-encoding every 10 steps, the average decoding error over the whole trajectory was 0.027, and the error at the $500^{\text{th}}$ step still remained as low as 0.037.

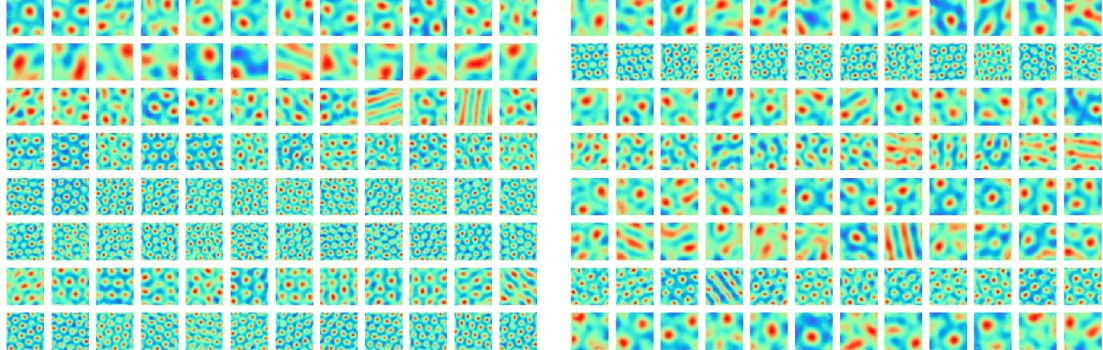

Figure 4: Learned hexagon grid patterns of $v(x)$, which is the hidden state vector in the LSTM transformation model. For each row, it shows all the cells in the same module, and 16 modules are randomly selected and visualized.

## 6. Conclusion and discussion

This paper investigates the algebraic, geometric, and topological properties of the generic transformation model of grid cells. In particular, we focus on non-linear recurrent neural networks under the conformal isometry condition. Our numerical experiments demonstrated that hexagon periodic patterns emerged in the response maps of grid cells under the conformal isometry condition. Our experiments also showed that the learned model was capable of accurate path integration.

The conformal isometry property seems to be related to the difference of Gaussian kernel assumed for place cells in Dordek et al. (2016); Sorscher et al. (2019), which constrains the frequency components within a ring in the frequency domain. It remains to be determined whether the hexagon patterns of grid cells were caused by the recurrent network itself or by interaction with place cells, an important issue that should be investigated further.

A limitation of our work is the lack of specially designed recurrent neural networks where conformal isometry is automatically satisfied. We leave it to future investigation. In the Appendix C, we provide a discussion of a possible design inspired by the hand-crafted continuous attractor neural networks of grid cells where conformal isometry is satisfied.

## Acknowledgments

The work was supported by NSF DMS-2015577 and XSEDE grant ASC170063. We thank the reviewers for their valuable comments and suggestions.

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

## Appendix A. More experimental details

### A.1. Training details

We train the model for $200,000$ iterations and learn the model by minimizing $L_1 + \lambda L_2$, where $\lambda = 0.05$. For the extra zero-step version of $L_1$, the weight is set as 10. The regularization of $\|\boldsymbol{q}(\boldsymbol{p})\|^2$ is added to the loss in the first $10,000$ iterations, where we linearly decay the weight of the regularization from 0.1 to 0. During training, the normalization of $\|\boldsymbol{v}(\boldsymbol{x})\|$ is done for every position $\boldsymbol{x}$ at the end of each epoch. For each module, we normalize the value of $\|\boldsymbol{v}(\boldsymbol{x})\|$ to be $1/\sqrt{\boldsymbol{m}}$, where $\boldsymbol{m}$ is the number of modules. For isometry loss ($L_2$), we fix the upper bounds of displacements as 15 grids for all modules for the first $10,000$ iterations, and start to adaptively adjust the upper bounds afterwards every 2000 iterations. All the learned parameters are updated by Adam (Kingma and Ba, 2014) optimizer. The learning rate is linearly decayed from 0.006 to 0.0003 for the first $10,000$ iterations, fixed at 0.0003 until $120,000$ iterations, and then linearly decayed to 0 afterwards. For batch sizes, we use 8000 for zero-step transformation loss and isometry loss ($L_1$), and 100 for multi-step transformation loss. We trained all the models on a single 2080 Ti GPU.

### A.2. Learned patterns

In Figure 5, we show the autocorrelograms of the learned grid patterns from the vanilla recurrent network.

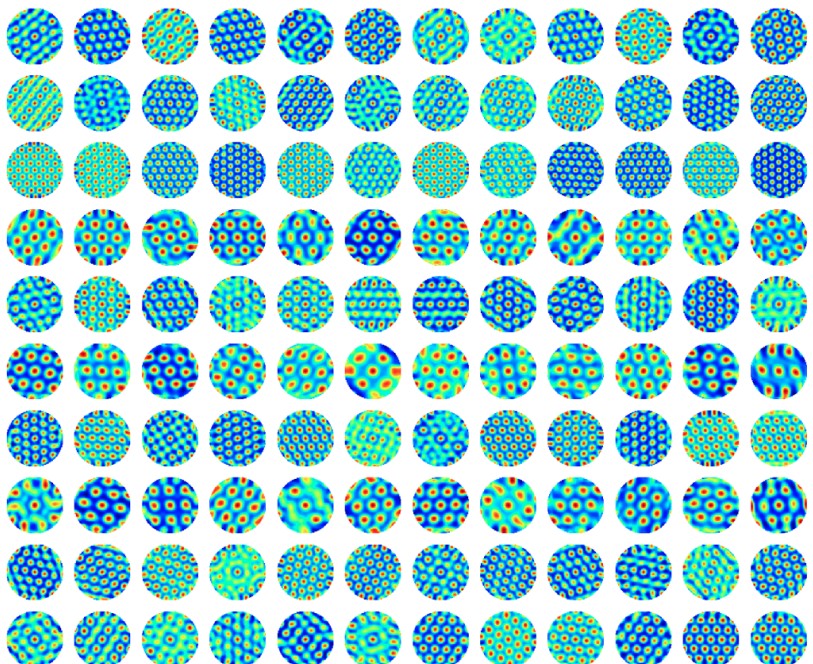

Figure 5: Autocorrelograms of the learned patterns.

### A.3. Ablation studies

In this section, we present part of our ablation results to examine whether certain components of our model are empirically important for the emergence of hexagon grid patterns. Here are some key observations: (1) hexagon patterns do not emerge without conformal isometry loss. That is, the conformal isometry condition is crucial for the emergence of grid-like patterns. (2) Regularization of $\|\boldsymbol{q}(\boldsymbol{p})\|^2$ is not necessary, but the patterns are less clear without it. (3) Without zero-step transformation loss, the learned model is unable to path integrate accurately, although hexagon grid patterns still emerge.

We further try different module sizes. Figure 8 visualizes the learned patterns when we fix the total number of grid cells and change the module size to 24. It shows hexagonal grid firing patterns can emerge with a larger module size. In Figure 9, we show the path integration error based on different lengths (5-step, 10-step, 15-step, and 30-step) of recurrent neural network models. Path integration for 30-step trajectories is performed for settings with and without re-encoding.

Finally, we try to remove the block-diagonal assumption in the transformation model, i.e. we change $\boldsymbol{W}$ to be a full matrix instead of a block-diagonal one. But we still impose conformal isometry on the modules. In this setting, we can still learn multi-scale patterns as in Figure 7, while the path integration also works well. For a learned 10-step vanilla RNN without block-diagonal construction, the average error of 500 step path integration is around 0.02 with re-encoding every 10 steps and 0.046 without re-encoding over the whole trajectory. This suggests that the emergence of the hexagon patterns is not from block-diagonal construction but the isometry condition.

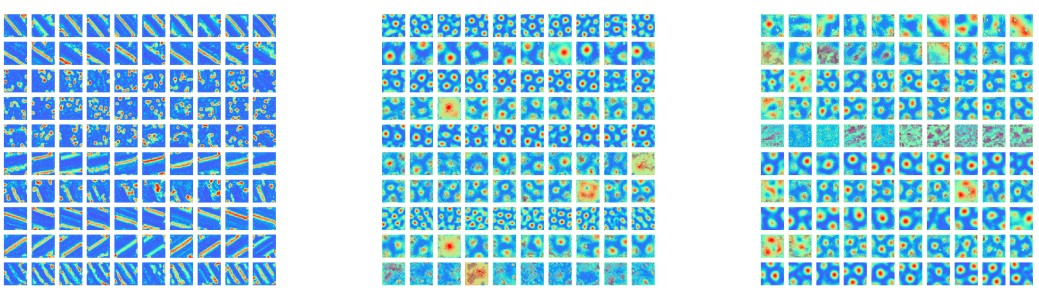

(a) Without isometry loss     (b) Without regularization of $\boldsymbol{u}$     (c) Without zero-step loss

Figure 6: Results of ablation on certain components of the training loss. (a) Learned patterns without conformal isometry loss. (b) Learned patterns without the regularization of $\|\boldsymbol{q}(\boldsymbol{p})\|^2$. (c) Learned patterns without zero-step transformation loss.

## Appendix B. Eigen analysis

**Proof** With $\Delta \boldsymbol{x} = 0$, we have the following fixed point property:

$$\boldsymbol{v}(\boldsymbol{x}) = \text{ReLU}(\boldsymbol{W}\boldsymbol{v}(\boldsymbol{x})), \tag{11}$$

$$\boldsymbol{v}(\boldsymbol{x} + \delta\boldsymbol{x}) = \text{ReLU}(\boldsymbol{W}\boldsymbol{v}(\boldsymbol{x} + \delta\boldsymbol{x})). \tag{12}$$

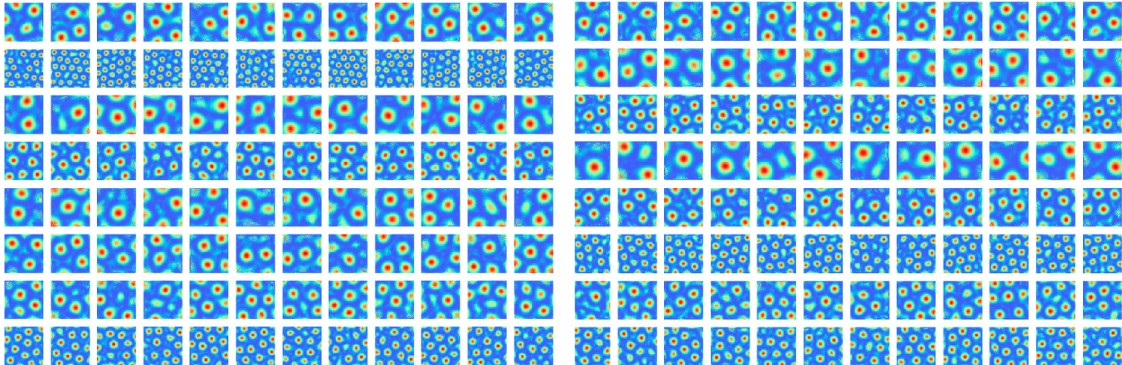

Figure 7: Hexagonal patterns emerged from the transformation model without block-diagonal assumption.

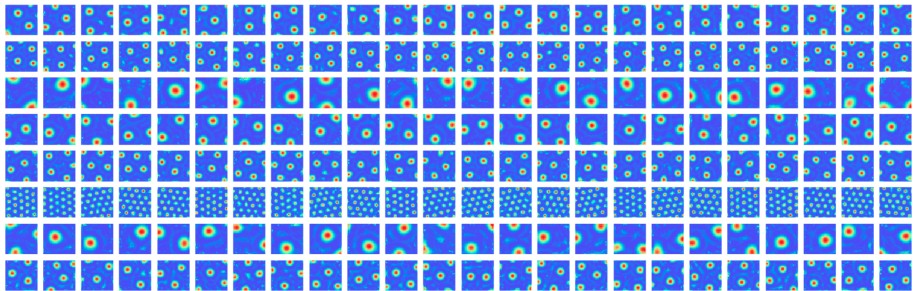

Figure 8: Learned patterns with module size 24.

Thus

$$\boldsymbol{v}(\boldsymbol{x} + \delta\boldsymbol{x}) - \boldsymbol{v}(\boldsymbol{x}) = \mathrm{ReLU}(\boldsymbol{W}\boldsymbol{v}(\boldsymbol{x} + \delta\boldsymbol{x})) - \mathrm{ReLU}(\boldsymbol{W}\boldsymbol{v}(\boldsymbol{x})) \tag{13}$$
$$= \mathbf{1}(\boldsymbol{W}\boldsymbol{v}(\boldsymbol{x}) > \mathbf{0}) \odot \boldsymbol{W}(\boldsymbol{v}(\boldsymbol{x} + \delta\boldsymbol{x}) - \boldsymbol{v}(\boldsymbol{x})) + o(|\delta\boldsymbol{x}|). \tag{14}$$

Thus for any $\boldsymbol{x}$, the matrix $\mathbf{1}(\boldsymbol{W}\boldsymbol{v}(\boldsymbol{x}) > \mathbf{0}) \odot \boldsymbol{W}$ has an eigenvalue 1 with geometric multiplicity 2. Meanwhile,

$$\boldsymbol{v}(\boldsymbol{x} + \delta\boldsymbol{x}) - \boldsymbol{v}(\boldsymbol{x}) = \mathrm{ReLU}(\boldsymbol{W}\boldsymbol{v}(\boldsymbol{x}) + \boldsymbol{U}\delta\boldsymbol{x}) - \mathrm{ReLU}(\boldsymbol{W}\boldsymbol{v}(\boldsymbol{x})) \tag{15}$$
$$= \mathbf{1}(\boldsymbol{W}\boldsymbol{v}(\boldsymbol{x}) > \mathbf{0}) \odot \boldsymbol{U}\delta\boldsymbol{x} + o(|\delta\boldsymbol{x}|). \tag{16}$$

Under conformal isometry, the two column vectors of $\mathbf{1}(\boldsymbol{W}\boldsymbol{v}(\boldsymbol{x}) > \mathbf{0}) \odot \boldsymbol{U}$ are orthogonal with equal norm, and they span the eigen-subspace of $\mathbf{1}(\boldsymbol{W}\boldsymbol{v}(\boldsymbol{x}) > \mathbf{0}) \odot \boldsymbol{W}$.

## Appendix C. Constructive conformal isometry

In our experiment, we impose conformal isomtry with a loss term based on Equation (4). In this section, we discuss a special case of the recurrent network that satisfies Condition 2 by design.

Specifically, we divide $\boldsymbol{v}$ into low-dimensional sub-vectors, $\boldsymbol{v} = (\boldsymbol{v}_k, k = 1, ..., K)$, where each $\boldsymbol{v}_k$ is a sub-vector of dimension $d$, e.g., $d = 4$ so that each sub-vector consists of 4

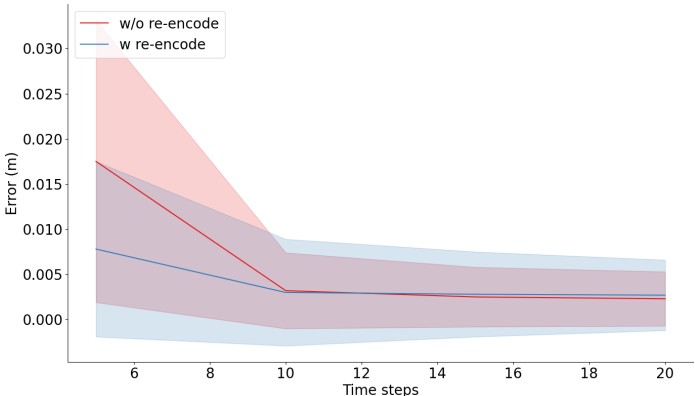

Figure 9: Path integration error over different lengths of RNN.

neurons. We call each sub-vector a mini-block. Then the total dimension $D = Kd$. Such mini-blocks are commonly assumed in handcrafted continuous attractor neural networks (Burak and Fiete, 2009; Couey et al., 2013; Pastoll et al., 2013; Agmon and Burak, 2020). Correspondingly, we divide the $D \times 2$ matrix $\boldsymbol{U}$ into $K$ mini-blocks, with each being $d \times 2$, so that $\boldsymbol{U} = (\boldsymbol{U}_k, k = 1, ..., K)$. The indicator vector $\mathbf{1}(\boldsymbol{W}\boldsymbol{v}(\boldsymbol{x}) > \mathbf{0})$ is also divided into $K$ mini-blocks, each of dimension $d$.

We assume the following two conditions for the mini-blocks:

**Condition 3.** *(Orthogonality condition). Each $\boldsymbol{U}_k$ has two column vectors that are of equal norm $s_k$ and are orthogonal to each other, i.e., $\boldsymbol{U}_k^\top \boldsymbol{U}_k = s_k^2 \boldsymbol{I}_2$.*

The above condition can be easily satisfied, e.g., for $d = 4$, we let the first column of $\boldsymbol{U}_k$ be $[1, 0, -1, 0]^\top$ and the second column be $[0, 1, 0, -1]^\top$. The idea is that within each mini-block, the cells are sensitive to different directions of self-motion.

**Condition 4.** *(Synchronicity condition). Each mini-block of $\mathbf{1}(\boldsymbol{W}\boldsymbol{v}(\boldsymbol{x}) > \mathbf{0})$ is synchronized, i.e., all its elements are either all 0 or all 1. For mini-block $k$, define $a_k = 1$ if all the elements of $\mathbf{1}(\boldsymbol{W}\boldsymbol{v}(\boldsymbol{x}) > \mathbf{0})$ are 1, and $a_k = 0$ if all the elements of $\mathbf{1}(\boldsymbol{W}\boldsymbol{v}(\boldsymbol{x}) > \mathbf{0})$ are 0.*

We call the mini-blocks that satisfy the above two conditions the conformal mini-blocks. Given those two conditions, we have the following result.

**Theorem 2.** *Under the orthogonality condition and the synchronicity condition, the mini-block-wise recurrent network satisfies the conformal isometry condition (Condition 2).*
**Proof** *The change of the vector*

$$\boldsymbol{v}(\boldsymbol{x} + \delta\boldsymbol{x}) - \boldsymbol{v}(\boldsymbol{x}) = \mathbf{1}(\boldsymbol{W}\boldsymbol{v}(\boldsymbol{x}) > \mathbf{0}) \odot \boldsymbol{U}\delta\boldsymbol{x} + o(|\delta\boldsymbol{x}|). \qquad (17)$$

*Thus*

$$\|\boldsymbol{v}(\boldsymbol{x} + \delta\boldsymbol{x}) - \boldsymbol{v}(\boldsymbol{x})\|^2 = \sum_k a_k s_k^2 \|\delta\boldsymbol{x}\|^2 + o(|\delta\boldsymbol{x}|^2) = s^2 \|\delta\boldsymbol{x}\|^2 + o(\|\delta\boldsymbol{x}\|^2), \qquad (18)$$

*where $s^2 = \sum_k a_k s_k^2$.*

Define the activation pattern of $\mathbf{1}(\boldsymbol{W}\boldsymbol{v}(\boldsymbol{x}) > \mathbf{0})$ be $\boldsymbol{a}(\boldsymbol{x}) = (a_k, k = 1, ..., K)$. $\boldsymbol{a}(\boldsymbol{x})$ controls the change of the vector $\boldsymbol{v}(\boldsymbol{x} + \delta\boldsymbol{x}) - \boldsymbol{v}(\boldsymbol{x})$. At different $\boldsymbol{x}$, $\boldsymbol{a}(\boldsymbol{x})$ are different. Thus the direction of the change $\boldsymbol{v}(\boldsymbol{x} + \delta\boldsymbol{x}) - \boldsymbol{v}(\boldsymbol{x})$ depends on $\boldsymbol{x}$, and it is possible for the model to create rotation of the vector $\boldsymbol{v}(\boldsymbol{x})$.

It is an interesting problem to construct simple recurrent networks that satisfy conformal isometry automatically.

