# OpenReview forum: "Conformal Isometry of Lie Group Representation in Recurrent Network of Grid Cells"
_NeurIPS.cc/2022/Workshop/NeurReps — NeurReps 2022 Poster_

### Official Review · Reviewer_UigM · 2022-10-11
**A reasonable little addition to grid cell theory**

**Confidence:** 4
**Soundness:** 3
**Presentation:** 2
**Contribution:** 2
**Overall Rating:** 5

**Summary:**

This work studies recurrent neural network models of the grid cell system. It shows that, when optimising the parameters of the network to minimise a few well-motivated loss terms, the optimal neural representation of space within the recurrent network bears resemblance to the famous modules of hexagonal grid cells.

This work is a small but interesting additional step relative to a very similar paper, Gao et al. 2021. In both papers the authors study a very similar set of losses: place cell decoding, path integration, conformal isometry, and some grid-place cell weight regularisation. In Gao et al. 2021 the velocity updated the representation via a velocity dependent matrix, in this work a couple of non-linear networks are used.

Another work, Sorscher, Mel, et al. 2020, (that should probably be cited in addition to, or in place of, Sorscher et al. 2019, as it includes further work) pursued a similar approach, training an RNN on a collection of losses and observed the emergence of hexagonal grids. The major difference seems to be that this work uses gaussian place cells and a conformal isometry loss, while Sorscher, Mel et al. use Mexican-hat place cells and no conformal isometry. Since place cells are not Mexican-hat like it is good to see another alternative!


**Questions:**

My main set of questions/concerns were about the modules:

Without enforcing the modularity of the transitions to begin with, could this scheme learn modules?

In the experiments section it says each displacement is smaller than three grids, what is a grid? Further, it seems that each module might only learn a different lengthscale because of the size of the steps used to enforce the conformal isometry loss, $\Delta r$, have I misunderstood that? If all grids were trained with shifts of the same length would there be a diversity in module sizes? It also seems that the maximum size of the module lengthscale could be set by the largest $\Delta r$, is this true?

Finally on modules: 150 is a lot! Why so many, I’ve heard there are only 7 in rats? And figure 7 when there are 24 neurons per module it felt like some of the grids were a little less hexagonal (though still impressively clean!) What happens if you have only 7 modules each with 100 neurons, or something like that?

This set of questions seems important since figure 2 claims there is a correspondence between this work and the rat findings, and it is very unclear to me which aspects of the numerics of the scheme were necessary for this to be true! If there is only correspondence for modules with very few neurons that would make an interesting prediction about connectivity in the brain?

And a couple of small comments:

I was confused by the relevance of paragraph 2 in 2.3, grid cells like on a torus, but not a flat torus. Was this paragraph just explaining that a perfectly conformal representation would have to be a flat torus? And that grid cells are not a flat torus for other reasons?

Paragraph 3 in 2.3, there are many (infinitely?) types of lattices beyond square and hexagonal in 2D, no?! Just squash a square slightly?

Again paragraph 3 of 2.3, I did not understand the claim about better fit of place cells to grid cells, what did this mean?

In section 5.3, what was u? Had it been introduced?


**Limitations:**

The authors do, to their credit, clearly discuss limitations of their work.

**Recommended Decision:**

2: Borderline

**Relevance:**

4: Highly relevant

**Strengths And Weaknesses:**

Originality:

The work is novel, taking a natural additional step from earlier work. This step is slightly clarifying, though not a revelation.

Quality:

The submission seems technically sound, the maths was clear, and the optimisation seemed well done. All claims had supporting evidence. I had concerns about the interpretation of some of the results, mainly those surrounding figure 2, as I discuss in my questions below.


Clarity:

The submission was mostly clear. There were times when I wasn't clear why something was being introduced, more way marking could help (the beginning of section 2.2, for example, could have begun 'we will now introduce the conformal loss yada yada yada'), but everything to understand the material was included.

Significance:

The main novel results was the optimisation which seemed interesting. This optimisation was a combination of ideas in previous literature, so most of the paper was running through these ideas.



**Submission Track:**

Proceedings Paper (9 Page)

---

### Official Review · Reviewer_hRfz · 2022-10-11
**CANs with conformal isometry**

**Confidence:** 4
**Soundness:** 2
**Presentation:** 2
**Contribution:** 2
**Overall Rating:** 5

**Summary:**

This paper build of Gao et al., 2021. In particular it exchanges Gao et al.'s lienar transformation model (inspired by group theory) with a CAN. Thus they show that a CAN that predict place cell representations learns lovely looking grid cells with the conformal isometry condition.

**Questions:**

The crucial question is whether conformal isometry is actually a property of grid cells? Would be well worth testing... It puts very specific constraints on the form of the grid code. Since grid cells are periodic then the situation can arise where, between two distant points, only the low frequency grid module differ - thus for this to equate to a large distance either the activity of low frequency grid cells needs to be high, or there need to be lots of them. Neither of these thigns are true in the data in my knowledge.

**Limitations:**

To make this really interesting, you would need to do a few things. 1) Test the isometry point in data. 2) Make the mathematical anayses really tell you something about what the weights in a CANs should look like, or what the manifolds of a CAN could look like.

**Recommended Decision:**

3: Accept

**Relevance:**

3: Solid fit

**Strengths And Weaknesses:**

Originality: This is an extention of Gao et al., 2021, to use a CAN update model on thie grids rathen than a linear tranformation mdoel.
Quality: Get very pretty grid cells, but the mathmatical analysis isn't very revealing.
Clarity: Good.
Significance: This is of some interest to the literature.

**Submission Track:**

Proceedings Paper (9 Page)

---

### Official Review · Reviewer_mW64 · 2022-10-13
**A paper exploring how adding a conformal isometry term to the loss function affects hidden representations.**

**Confidence:** 5
**Soundness:** 3
**Presentation:** 3
**Contribution:** 2
**Overall Rating:** 5

**Summary:**

This paper explores how recurrent network models, trained to output an idealized place cell-like representation of 2D space can produce grid-like patterns in their hidden representation with velocity inputs and conformal isometry term in the loss.

**Questions:**

In general, when claiming that certain representations in artificial networks emerge, authors should always run important controls to make sure that these representations emerge robustly and are not a consequence of appropriately chosen post-hoc hyper-parameter choices.

A few experiments or analyses the authors should in my opinion try or discuss are:

- What happens when the network is not initialized in the block-diagonal-like manner?

- What happens to the hidden representation if the encoding of 2D space is not by idealized, single scale and isotropic place cells but rather by another function like (x,y) or (r, $\theta$) or if the place cells are more naturalistic as in the experiments of Shaeffer et al (heterogenous scales, multi-bumped)? Under these conditions, it was shown that grid-like representations do not emerge and hence their emergence is more a consequence of the choice of cleverly chosen supervised target encoding rather than fundamental constraints or properties imposed by the task of path integration or loss function.

**Limitations:**

As suggested before, including a section where the authors would comment on the relation of the present work to that of Shaeffer et al [1] and discussing whether their work would face the same pitfalls that were pointed out by [1], would be very useful to the community.

[1] https://openreview.net/forum?id=mxi1xKzNFrb

**Recommended Decision:**

3: Accept

**Relevance:**

3: Solid fit

**Strengths And Weaknesses:**

Strengths:
- The inclusion of the conformal isometry loss is well motivated from group theoretic properties and seems to result in clean grid cells. This loss function is general, and I can see it being useful to the NeurReps community beyond only studying grid-like representations.

There are a few weaknesses that the authors should address and that need to be pointed out.

- To build multiple modules, the authors used a block-diagonal construction on their weight matrices. Other works (for example: Banino et al and Sorscher et al) that claim to produce grid cells from a path-integrating RNN that outputs place cell representations usually get multiple modules without such a construction. The authors have not discussed or reported what happens when the block diagonal constraint is removed. It is also not clear what exactly is the difference between each block, so this assumption is hard to evaluate.

- This paper is very incremental over the previous work cited: Gao et al 2021. The only difference seems to be the use of a non-linear updates (an LSTM) and the replacement of the 'isotropic scaling' term of the loss function of Gao et al 2021 with the conformal isometry term.

- Recent work (Shaeffer et al[1]) has shown that path integrating RNNs that output place cell representations of 2D space are not genuine models that produce grid cells. Upon changing the nature of encoding of 2D space, grid cell-like representations vanish. The authors should add a discussion of their work and comment on whether it would face the same limitations as pointed out by [1].

[1] Shaeffer et al: https://openreview.net/forum?id=mxi1xKzNFrb

**Submission Track:**

Proceedings Paper (9 Page)

---

### Decision · Program_Chairs · 2022-10-21

Accept (Poster)